# Mask Propagation for Efficient Video Semantic Segmentation

**Yuetian Weng**[1,2*] **Mingfei Han**[3,4] **Haoyu He**[1] **Mingjie Li**[3]
**Lina Yao**[4] **Xiaojun Chang**[3,5] **Bohan Zhuang**[1†]
[1]ZIP Lab, Monash University    [2]Baidu Inc.    [3]ReLER, AAII, UTS
[4]Data61, CSIRO    [5]Mohamed bin Zayed University of AI

## Abstract

Video Semantic Segmentation (VSS) involves assigning a semantic label to each pixel in a video sequence. Prior work in this field has demonstrated promising results by extending image semantic segmentation models to exploit temporal relationships across video frames; however, these approaches often incur significant computational costs. In this paper, we propose an efficient mask propagation framework for VSS, called `MPVSS`. Our approach first employs a strong query-based image segmentor on sparse key frames to generate accurate binary masks and class predictions. We then design a flow estimation module utilizing the learned queries to generate a set of segment-aware flow maps, each associated with a mask prediction from the key frame. Finally, the mask-flow pairs are warped to serve as the mask predictions for the non-key frames. By reusing predictions from key frames, we circumvent the need to process a large volume of video frames individually with resource-intensive segmentors, alleviating temporal redundancy and significantly reducing computational costs. Extensive experiments on VSPW and Cityscapes demonstrate that our mask propagation framework achieves SOTA accuracy and efficiency trade-offs. For instance, our best model with Swin-L backbone outperforms the SOTA MRCFA using MiT-B5 by 4.0% mIoU, requiring only 26% FLOPs on the VSPW dataset. Moreover, our framework reduces up to $4\times$ FLOPs compared to the per-frame Mask2Former baseline with only up to 2% mIoU degradation on the Cityscapes validation set. Code is available at https://github.com/ziplab/MPVSS.

## 1   Introduction

Video Semantic Segmentation (VSS), a fundamental task in computer vision, seeks to assign a semantic category label to each pixel in a video sequence. Previous research on VSS has leveraged developments in image semantic segmentation models, *e.g.*, FCN [40] and Deeplab [82, 4], which made tremendous progress in the field. However, adapting image semantic segmentation models to VSS remains challenging. On one hand, sophisticated temporal modeling is required to capture the intricate dynamics among video frames. On the other hand, videos contain a significantly larger volume of data compared to images. Processing every frame with a strong image segmentor can incur significant computational costs.

Prior work in VSS mainly focuses on leveraging both temporal and spatial information to enhance the accuracy of pixel-wise labeling on each video frame, building upon the pixel-wise classification paradigm of traditional image semantic segmentors. Specifically, these methods exploit temporal

---

*Work done during an internship at Baidu Inc.

†Corresponding author. Email: `bohan.zhuang@gmail.com`

37th Conference on Neural Information Processing Systems (NeurIPS 2023).

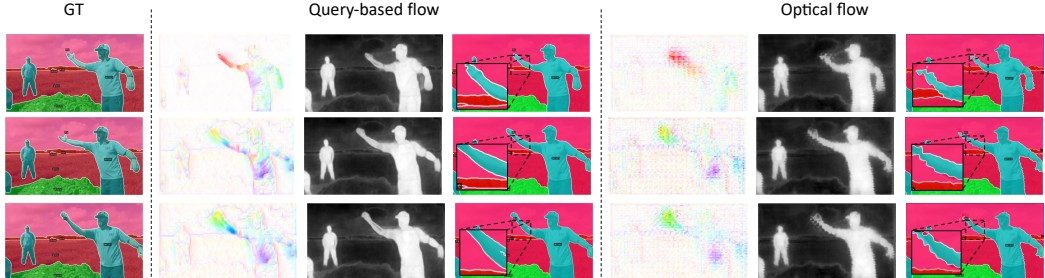

Figure 1: Motivation of the proposed MPVSS. Three consecutive frames with ground truth category labels are sampled from a video in VSPW [42], illustrating a strong correlation between video frames along the temporal dimension. This observation underscores the significant temporal redundancy present in videos. The remaining columns contrast the proposed query-based flow and traditional optical flow, each showing a normalized flow map, a mask prediction, and the final predicted semantic map, respectively.

information by integrating visual features from previous frames into the target frame, employing techniques such as recurrent units [26, 43], temporal shift [33, 1], or spatial-temporal attention modules [19, 44, 28, 16, 53], among others.

More recently, with the prevalence of the DETR [2] framework, DETR-like segmentors [8, 7, 51] have dominated the latest advances and achieved state-of-the-art performance in semantic segmentation. Notably, MaskFormer series [8, 7] learn a set of queries representing target segments, and employ bipartite matching with ground truth segments as the training objective. Each query is learned to predict a binary mask and its associated class prediction. The final result is obtained by combining the binary masks and class predictions of all queries via matrix multiplication. However, these models require computing a high-resolution per-pixel embedding for each video frame using a powerful image encoder, leading to significant computational costs. For instance, processing an input video clip with $30 \times 1024 \times 2048$ RGB frames using the strong Mask2Former [7] requires more than 20.46T Floating-point Operations (FLOPs) in total. Such computational demands render it impractical to deploy these models in real-world VSS systems.

A typical solution is to incorporate temporal information guided by optical flow [50, 82, 22, 66] to reduce the redundancy. The related literature proposes to propagate feature maps from the key frame to other non-key frames using optical flow. The motivation mainly comes from two aspects. First, videos exhibit high temporal redundancy due to the similar appearance of objects or scenes in consecutive frames [13], especially in videos with a high frame rate. As shown in Fig. 1, we empirically observe that semantic information in consecutive video frames is highly correlated. Additionally, previous methods [50, 82, 22, 30, 25, 45] highlight the gradual changes in semantic features within deep neural networks. They attempt to reduce the computational costs by reusing feature maps from preceding frames for the target frame, facilitated by the estimated optical flow between the neighboring frames. However, these methods may still suffer from degraded performance due to inaccurate and noisy optical flow estimation, which only measures pixel-to-pixel correspondence between adjacent frames. Recent advancements in Transformer-based optical flow methods [58, 20, 63, 64] demonstrate the importance of incorporating global correspondence information for each pixel, resulting in more precise per-pixel optical flow estimation. Nevertheless, these methods still focus solely on pixel-level correspondence and do not account for segment-level information among video frames, as illustrated in Fig. 1, which may be insufficient for the task of VSS.

In this paper, we present an efficient *mask propagation* framework for VSS, namely MPVSS, as shown in Fig. 2. This framework relies on two key components: *a strong query-based image segmentor* for the key frame (upper part of Fig. 2), and *a powerful query-based flow estimator* for the non-key frames (lower part of Fig. 2). Specifically, we employ the state-of-the-art query-based image segmentation model, *i.e.*, Mask2Former [7] to process sparse key frames and generate accurate binary masks and class predictions. We then propose to estimate individual flow map corresponding to each segment-level mask prediction of the key frame. To achieve this, we leverage the learned queries from the key frame to aggregate segment-level motion information between adjacent frame pairs and capture segment-specific movements along the temporal dimension. These queries are fed to a flow head to predict a set of segment-aware flow maps, followed by a refinement step. The mask

predictions from key frames are subsequently warped to other non-key frames through the generated query-based flow maps, where each flow map is tailored to the corresponding mask predicted by the associated query in the key frame. Finally, we derive semantic maps for the non-key frames by aggregating the warped mask predictions with the class probabilities predicted from the key frame. By warping the predicted segmentation masks from key frames, our model avoids processing each individual video frame using computationally intensive semantic segmentation models. This not only reduces temporal redundancy but also significantly lowers the computational costs involved.

In summary, our main contributions are in three folds:

- We propose MPVSS, a novel mask propagation framework for efficient VSS, which reduces computational costs and redundancy by propagating accurate mask predictions from key frames to non-key frames.

- We devise a novel query-based flow estimation module that leverages the learned queries from the key frame to model motion cues for each segment-level mask prediction, yielding a collection of accurate flow maps. These flow maps are utilized to propagate mask predictions from the key frame to other non-key frames.

- Extensive experiments on standard benchmarks, VSPW [42] and Cityscapes [9], demonstrate that our MPVSS achieves SOTA accuracy and efficiency trade-offs.

## 2   Related Work

**Image semantic segmentation.** As a fundamental task in computer vision, image semantic segmentation seeks to assign a semantic label to each pixel in an image. The pioneering work of Fully Convolutional Networks (FCNs) [40] first adopts fully convolutional networks to perform pixel-wise classification in an end-to-end manner and apply a classification loss to each output pixel, which naturally groups pixels in an image into regions of different categories. Building upon the per-pixel classification formulation, various segmentation methods have been proposed in the past few years. Some works aim at learning representative features, which propose to use atrous convolutional layers to enlarge the receptive field [68, 4, 5, 46, 67], aggregate contextual information from multi-scale feature maps via a pyramid architecture [17, 36], encoder-decoder architecture [49, 6], or utilize attention modules to capture the global dependencies [27, 76, 14, 79, 78, 61]. Another line of work focuses on introducing boundary information to improve prediction accuracy for details [10, 29, 70, 77]. More recent methods demonstrate the effectiveness of Transformer-based architectures for semantic segmentation. SegFormer [62], Segmentor [51], SETR [78], MaskFormer [8, 7] replace traditional convolutional backbones [18] with Vision Transformers [11, 59, 39, 34] and/or implement the head with Transformers following the DETR-like framework [2]. Basically, we propose an efficient framework for video semantic segmentation that leverages the mask predictions generated by DETR-like image segmentors. Our framework focuses on propagating these mask predictions across video frames, enabling accurate and consistent segmentation results throughout the entire video sequence. By incorporating the strengths of DETR-like models in image segmentation, we achieve high-quality and efficient video semantic segmentation.

**Video semantic segmentation.** Video semantic segmentation has recently captured the attention of researchers due to its dynamic nature and practical significance in real-world scenarios. Compared with image semantic segmentation, most existing methods of VSS pay attention to exploiting temporal information, which can be divided into two groups. The first group of approaches exploits temporal relationships to improve prediction accuracy and consistency. For example, some works utilize recurrent units [26, 43] or design an attention propagation module [19] to warp the features or results of several reference frames [81, 15], or aggregate neighbouring frames based on optical flow estimation [15, 37]. In particular, ETC [38] proposes temporal consistency regularization during training, [73] instead exploits perceptual consistency in videos, MRCFA [54] mines relationship of cross-frame affinities, STT [28] employs spatial-temporal transformer, CFFM-VSS [53] disentangles static and dynamic context for video frames. The second line of methods infers to alleviate the huge temporal redundancy among video frames and thus reduce the computational cost. Observing the high similarity of the semantic representation of deep layers, ClockNet [50] designs a pipeline schedule to adaptively reuse feature maps from previous frames in certain network layers. DFF [82] utilizes optical flow to propagate feature maps from key frames to non-key frames, achieving remarkable computational efficiency. Accel [22] further proposes to execute a large model on key frames and

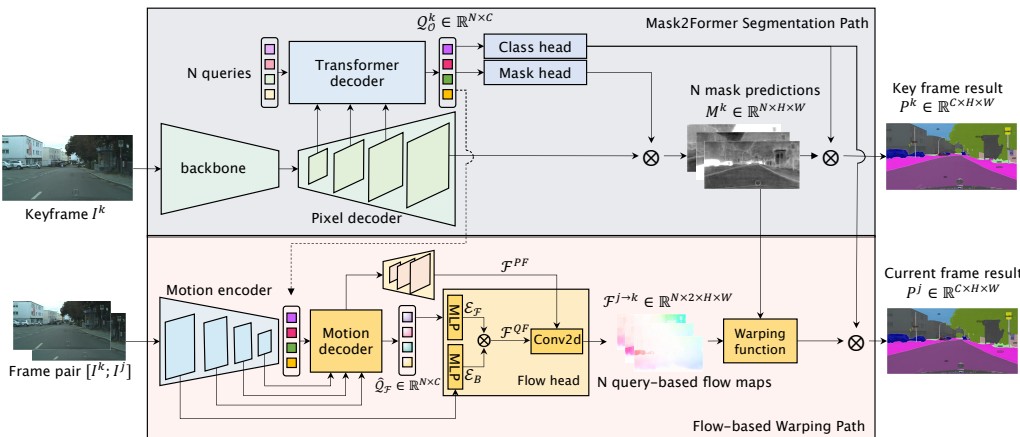

Figure 2: Overall architecture of the proposed MPVSS.

a compact one on non-key frames, while DVN [66] proposes a region-based execution scheme on non-key frames, as well as an dynamic key-frame scheduling policy. LLVS [30] develops an efficient framework involving both adaptive key frame selection and selective feature propagation. Unlike methods that primarily rely on optical flow, which models dense pixel-to-pixel correspondence, we introduce a novel query-based flow module that focuses on segment-level flow fields between adjacent frames. By operating at the segment level, our method captures higher-level motion information that is more accurate for VSS.

**Motion estimation by optical flow.** Optical flow estimation is a fundamental module in video analysis, which estimates a dense field of pixel displacement between adjacent frames. In deep learning era, FlowNet [12] first utilizes convolutional neural networks to learn from labeled data. Most follow-up methods propose to employ spatial pyramid networks [52], or utilize recurrent layers [58] to apply iterative refinement for the predicted flow in a coarse-to-fine manner [52, 47, 24, 55]. More recent methods pioneer Transformers to capture the global dependencies of cost volume [58, 20], or extract representative feature maps for global matching [63], addressing the challenge of large displacement. As optical flow represents the pixel-to-pixel correspondence in adjacent frames, it is often utilized in a wide range of video analysis tasks, *i.e.*, optical flow as motion information for modeling action motion in the tasks of action recognition [3, 56], temporal action detection [57, 32], or improving the feature representation in video segmentation [74], object detection [81]; optical flow estimation as a simultaneous task to model pixel correspondence for video interpolation [23, 48, 21], video inpainting [71, 72, 31], etc.

## 3 Mask Propagation via Query-based Flow

In Sec. 3.1, we introduce the overall pipeline of the proposed MPVSS. In Sec. 3.2, we simply revisit Mask2Former that MPVSS is built upon. Finally, in Sec. 3.3, we introduce our query-based flow estimation method, which yields a set of flow maps for efficiently propagating mask predictions from key frames to other non-key frames.

### 3.1 Overview

The overall architecture is shown in Fig. 2. Let $\{\mathbf{I}^t \in \mathbb{R}^{H \times W \times 3}\}_{t=1}^T$ be an input video clip with $T$ frames, where $H$ and $W$ denote the height and width of frames in the video. Without loss of generality, we select key frames from the video clip at fixed intervals, *e.g.*, every 5 frames, considering other frames within these intervals as non-key frames. For simplicity, we denote each key frame as $\mathbf{I}^k$ and non-key frame as $\mathbf{I}^j$. To reduce redundant computation, we only run the heavy image segmentation model on the sampled key frames and propagate the predictions from key frames to non-key frames.

Specifically, for a key frame $\mathbf{I}^k$, we adopt Mask2Former [7] (upper part of Fig. 2) to generate $N$ mask predictions $\mathbf{M}^k$ and class predictions. Then, we use the flow estimation module (bottom part

of Fig. 2) to predict a set of flow maps $\mathcal{F}^{j \to k} \in \mathbb{R}^{N \times 2 \times H \times W}$ indicating the motion changes from $\mathbf{I}^j$ to $\mathbf{I}^k$, which will be discussed in Sec. 3.3. Subsequently, we apply the bilinear warping function on all locations to propagate per-segment mask predictions $\mathbf{M}^k$ to obtain $\mathbf{M}^j$ of non-key frames, *i.e.*, $\mathbf{M}^j = \mathcal{W}(\mathbf{M}^k, \mathcal{F}^{j \to k})$. Consequently, we segment the non-key frame $\mathbf{I}^j$ by propagating the predicted masks from its preceding key frame $\mathbf{I}^k$, avoiding the necessity of computationally intensive image segmentation models for every video frame.

During training, we apply the loss function in Mask2Former [7] on the warped mask predictions $\mathbf{M}^j$ and perform bipartite matching with the ground truth masks for the current frame. The training approach is similar to the process described in [82], where the loss gradients are back-propagated throughout the model to update the proposed flow module.

## 3.2 Mask2Former for Segmenting Key Frames

Mask2Former consists of three components: a backbone, a pixel decoder and a transformer decoder, as depicted in the upper part of Fig. 2. The backbone extracts feature maps from the key frame $\mathbf{I}^k \in \mathbb{R}^{H \times W \times 3}$. The pixel decoder builds a multi-scale feature pyramid as well as generates a high-resolution per-pixel embedding, following FPN [35] and Deformable DETR [80]. In transformer decoder, the target segments are represented as a set of learned queries $\mathcal{Q}_{\mathcal{O}}^k \in \mathbb{R}^{N \times C}$ where $C$ is the channel dimension. After gradually enriching their representations with stacked decoder blocks, $\mathcal{Q}_{\mathcal{O}}^k$ is fed into a class and a mask head to yield $N$ class probabilities and $N$ mask embeddings, respectively. After decoding the binary mask predictions $\mathbf{M}^k \in \mathbb{R}^{N \times H \times W}$ with the class predictions and the per-pixel embeddings, we can finally derive the semantic maps for the key frame by aggregating $N$ binary mask predictions with their corresponding predicted class probabilities via matrix multiplication. We refer readers to [8, 7] for more details.

## 3.3 Query-based Flow Estimation

To generate the segmentation masks for the non-key frames efficiently, we propose to estimate a set of flow maps, each corresponding to a mask prediction of the key frame. We start with introducing the process of generating $N$ query-based flow maps between the current non-key frame $\mathbf{I}^j$ and its preceding key frame $\mathbf{I}^k$, where $0 < j - k \leq T$. The overall flow module comprises three key elements: a motion encoder to encode the motion feature pyramid between the pair of video frames $\{\mathbf{I}^k, \mathbf{I}^j\}$, a motion decoder that leverages the learned queries $\mathcal{Q}_{\mathcal{O}}^k$ from the key frame to extract motion information from the motion feature maps for each segment, and a flow head responsible for predicting $N$ query-based flow maps $\mathcal{F}^{j \to k}$, where each flow map is associated with warping a segment (mask prediction) of the key frame to the current non-key frame.

**Motion encoder.** To obtain motion features between the paired frames $\{\mathbf{I}^k, \mathbf{I}^j\}$, we concatenate the two frames along the channel dimension as the input of the motion encoder. The motion encoder, utilizing the same architecture as the flow encoder in FlowNet [12], is employed to extract motion features at each location in a downsampling manner. Following [12, 7], we reverse the order of the original feature maps, generating a motion feature pyramid $\{\mathbf{b}_1, \mathbf{b}_2, \mathbf{b}_3, \mathbf{b}_4\}$, each at resolutions $1/32$, $1/16$, $1/8$ and $1/4$ of the original video frame, respectively. Subsequently, the feature pyramid is fed to the model decoder for the decoding of flow maps from the lowest to the highest resolution.

**Motion decoder.** We then devise a motion decoder to aggregate motion information from the motion feature pyramid using $N$ flow queries $\mathcal{Q}_{\mathcal{F}}$.

First, the motion decoder takes as input the first three feature maps $\{\mathbf{b}_1, \mathbf{b}_2, \mathbf{b}_3\}$ from the motion feature pyramid and the flow queries $\mathcal{Q}_{\mathcal{F}}$. We initialize flow queries with the $N$ learned queries $\mathcal{Q}_{\mathcal{O}}^k$ of the preceding key frame, *i.e.*, $\mathcal{Q}_{\mathcal{F}}^1 = \mathcal{Q}_{\mathcal{O}}^k$, where $\mathcal{Q}_{\mathcal{F}}^1$ indicates the input flow queries to the motion decoder. Following [7], feature map in each layer is projected into $C$ channels in the channel dimension using a linear layer, which is then added with a sinusoidal positional embedding and a learnable level embedding. This results in three scales of motion feature maps in total, *i.e.*, $\{\mathbf{B}_1^1, \mathbf{B}_2^1, \mathbf{B}_3^1\}$, as the input of the following layers.

Next, the motion decoder can be divided into $S$ stages, each indexed by $s$, with each stage involving $L$ blocks, each indexed by $l$. We adopt $S = 3$ and $L = 3$ for our model. Inspired by [7, 65], starting from $s = 1$ and $l = 1$, for each block in each stage, we first concatenate $\mathbf{B}_l^s$ and $\mathcal{Q}_{\mathcal{F}}^s$ together and then feed them into a number of Transformer layers, each of which performs information propagation

between $\mathbf{B}_l^s$ and $\mathcal{Q}_{\mathcal{F}}^s$.

$$\mathbf{Z}_l^s = [\mathcal{Q}_{\mathcal{F}}^s; \mathbf{B}_l^s], \tag{1}$$

$$\hat{\mathbf{Z}}_l^s = \text{LN}(\text{MSA}(\mathbf{Z}_l^s) + \mathbf{Z}_l^s), \tag{2}$$

$$\hat{\mathcal{Q}}_{\mathcal{F}}^s, \hat{\mathbf{B}}_l^s = \text{LN}(\text{FFN}(\hat{\mathbf{Z}}_l^s) + \hat{\mathbf{Z}}_l^s), \tag{3}$$

where $[;]$ denotes the concatenation operation. $\text{LN}(\cdot)$, $\text{MSA}(\cdot)$ and $\text{FFN}(\cdot)$ denote LayerNorm layer, Mutli-head Self-Attention and Feed-forward Neural Network, respectively. Therefore, in each stage, we learn flow queries and update motion feature maps via self-attention. The self-attention mechanism aggregates motion information globally from both the motion feature maps and the flow query features simultaneously. Finally, the learned flow queries $\hat{\mathcal{Q}}_{\mathcal{F}}$ and updated motion feature maps $\{\hat{\mathbf{B}}_1, \hat{\mathbf{B}}_2, \hat{\mathbf{B}}_3\}$ are fed to the flow head for flow prediction.

As each query of $\mathcal{Q}_{\mathcal{O}}^k$ encodes global information for each segment in the key frame, using $\mathcal{Q}_{\mathcal{O}}^k$ as initial flow queries enables each flow query to extract motion information for each segment present in the key frame, which facilitates query-based motion modeling.

**Flow head.** The flow head generate the final flow maps $\mathcal{F}^{j \rightarrow k} \in \mathbb{R}^{N \times 2 \times H \times W}$ by combining query-based flow $\mathcal{F}^{QF}$ and pixel-wise flow $\mathcal{F}^{PF}$.

To obtain the query-based flow $\mathcal{F}^{QF}$, an MLP with 2 hidden layers is employed to convert the learnt flow queries $\hat{\mathcal{Q}}_{\mathcal{F}}$ to flow embeddings $\mathcal{E}_{\mathcal{F}} \in \mathbb{R}^{N \times C}$, each with $C$ channels. Inspired by the pixel decoder design in Mask2Former, we project the motion feature map $\mathbf{b}_4$ from the motion encoder to $\mathcal{E}_B \in \mathbb{R}^{2C \times \frac{H}{4} \times \frac{W}{4}}$ using another MLP with 2 hidden layers. Let $\mathcal{E}_B^x$ and $\mathcal{E}_B^y$ denote the first and last $C$ channels of $\mathcal{E}_B$, and they are used to obtain the query-based flow predictions in terms of $x$ and $y$ direction, respectively. Then, for the $n^{th}$ query, we get the flow map via a dot product between the $n^{th}$ flow embedding $\mathcal{E}_{\mathcal{F}_n}$ and the projected motion map $\mathcal{E}_B$, *i.e.*, $\mathcal{F}_n^{QF}(x) = \mathcal{E}_{\mathcal{F}_n} \cdot \mathcal{E}_B^x$ and $\mathcal{F}_n^{QF}(y) = \mathcal{E}_{\mathcal{F}_n} \cdot \mathcal{E}_B^y$ in terms of $x$ and $y$ direction, respectively. Then we can obtain the predicted query-based flow $\mathcal{F}_n^{QF} = [\mathcal{F}_n^{QF}(x); \mathcal{F}_n^{QF}(y)] \in \mathbb{R}^{2 \times \frac{H}{4} \times \frac{W}{4}}$ for the $n^{th}$ query. The dot product of the flow embeddings and projected motion maps quantifies the similarity between the segment-level motion vector and the encoded motion information at each location on the feature maps, which reflects the movement for each query on the spatial dimension.

Additionally, we use the updated motion feature maps $\{\hat{\mathbf{B}}_1, \hat{\mathbf{B}}_2, \hat{\mathbf{B}}_3\}$ to predict a pixel-wise flow $\mathcal{F}^{PF} \in \mathbb{R}^{2 \times \frac{H}{4} \times \frac{W}{4}}$ in a coarse-to-fine manner, following [12], which is used to refine the query-based flow.

To obtain the final flow map $\mathcal{F}_n^{j \rightarrow k} \in \mathbb{R}^{2 \times H \times W}$ for the $n^{th}$ query, we apply a 2D convolutional layer to the concatenated $\mathcal{F}_n^{QF}$ and $\mathcal{F}^{PF}$, and then upsample the flow map to the original resolution:

$$\mathcal{F}_n^{j \rightarrow k} = \text{Upsample}\left(\text{Conv2D}\left([\mathcal{F}_n^{QF}; \mathcal{F}^{PF}]\right)\right). \tag{4}$$

**Remark.** We use $N$ flow queries, which are initialized with the learned per-segment queries $\mathcal{Q}_{\mathcal{O}}^k$ from the key frame, to capture the motion for each mask predicted on the key frame. In contrast to existing optical flow estimation methods, which model pixel-to-pixel dense correspondence, we propose a novel query-based flow that incorporates segment-level motion information between adjacent frames, assisting more accurate mask propagation in VSS.

## 4 Experiments

### 4.1 Experimental Setup

**Datasets.** We evaluate our method on two benchmark datasets: VSPW [42] and Cityscapes [9]. VSPW is the largest video semantic segmentation benchmark, consisting of 2,806 training clips (198,244 frames), 343 validation clips (24,502 frames), and 387 test clips (28,887 frames). Each video in VSPW has densely annotated frames at a rate of 15 frames per second, covering 124 semantic categories across both indoor and outdoor scenes. Additionally, we present results on the Cityscapes dataset, which provides annotations every 30 frames. This dataset serves as an additional evaluation to showcase the effectiveness of our proposed method.

Table 1: Performance comparisons with state-of-the-art methods on VSPW dataset. We report mean IoU (mIoU) and Weighted IoU (WIoU) for the performance evaluation and Video Consistency (VC) for the temporal consistency comparison. We measure FLOPs (G) for computational costs, averaging over a 15-frame clip with resolution of $480 \times 853$. Frame-per-second (FPS) is measured on a single NVIDIA V100 GPU with 3 repeated runs.

| Methods | Backbone | mIoU↑ | WIoU↑ | $mVC_8$ ↑ | $mVC_{16}$ ↑ | GFLOPs ↓ | Params(M)↓ | FPS↑ |
|---|---|---|---|---|---|---|---|---|
| DeepLab3+ [4] | R101 | 34.7 | 58.8 | 83.2 | 78.2 | 379.0 | 62.7 | 9.25 |
| UperNet [60] | R101 | 36.5 | 58.6 | 82.6 | 76.1 | 403.6 | 83.2 | 16.05 |
| PSPNet [75] | R101 | 36.5 | 58.1 | 84.2 | 79.6 | 401.8 | 70.5 | 13.84 |
| OCRNet [69] | R101 | 36.7 | 59.2 | 84.0 | 79.0 | 361.7 | 58.1 | 14.39 |
| ETC [38] | OCRNet | 37.5 | 59.1 | 84.1 | 79.1 | 361.7 | - | - |
| NetWarp [15] | OCRNet | 37.5 | 58.9 | 84.0 | 79.0 | 1207 | - | - |
| TCB [42] | R101 | 37.8 | 59.5 | 87.9 | 84.0 | 1692 | - | - |
| Segformer [62] | MiT-B2 | 43.9 | 63.7 | 86.0 | 81.2 | 100.8 | 24.8 | 16.16 |
| Segformer | MiT-B5 | 48.9 | 65.1 | 87.8 | 83.7 | 185.0 | 82.1 | 9.48 |
| CFFM-VSS [53] | MiT-B2 | 44.9 | 64.9 | 89.8 | 85.8 | 143.2 | 26.5 | 10.08 |
| CFFM-VSS | MiT-B5 | 49.3 | 65.8 | 90.8 | 87.1 | 413.5 | 85.5 | 4.58 |
| MRCFA [54] | MiT-B2 | 45.3 | 64.7 | 90.3 | 86.2 | 127.9 | 27.3 | 10.7 |
| MRCFA | MiT-B5 | 49.9 | 66.0 | 90.9 | 87.4 | 373.0 | 84.5 | 5.02 |
| Mask2Former [7] | R50 | 38.5 | 60.2 | 81.3 | 76.4 | 110.6 | 44.0 | 19.4 |
| Mask2Former | R101 | 39.3 | 60.1 | 82.5 | 77.6 | 141.3 | 63.0 | 16.90 |
| Mask2Former | Swin-T | 41.2 | 62.6 | 84.5 | 80.0 | 114.4 | 47.4 | 17.13 |
| Mask2Former | Swin-S | 42.1 | 63.1 | 84.7 | 79.3 | 152.2 | 68.9 | 14.52 |
| Mask2Former | Swin-B | 54.1 | 70.3 | 86.6 | 82.9 | 223.5 | 107.1 | 11.45 |
| Mask2Former | Swin-L | 56.1 | 70.8 | 87.6 | 84.1 | 402.7 | 215.1 | 8.41 |
| MPVSS | R50 | 37.5 | 59.0 | 84.1 | 77.2 | 38.9 | 84.1 | 33.93 |
| MPVSS | R101 | 38.8 | 59.0 | 84.8 | 79.6 | 45.1 | 103.1 | 32.38 |
| MPVSS | Swin-T | 39.9 | 62.0 | 85.9 | 80.4 | 39.7 | 114.0 | 32.86 |
| MPVSS | Swin-S | 40.4 | 62.0 | 86.0 | 80.7 | 47.3 | 108.0 | 30.61 |
| MPVSS | Swin-B | 52.6 | 68.4 | 89.5 | 85.9 | 61.5 | 147.0 | 27.38 |
| MPVSS | Swin-L | 53.9 | 69.1 | 89.6 | 85.8 | 97.3 | 255.4 | 23.22 |

**Implementation details.** By default, all experiments are trained with a batch size of 16 on 8 NVIDIA GPUs. Note that pre-training is important for the DETR-like segmentors. For VSPW dataset, we pretrain Mask2Former on the image semantic segmentation task, serving as the per-frame baseline. For Cityscapes, we adopt pretrained Mask2Former with ResNet [18] and Swin Transformer [39] backbones from [7]. For the motion encoder, we utilized the weights from FlowNet [12] encoder which is pre-trained on the synthetic Flying Chairs dataset; the motion decoder and flow head are randomly initialized. We freeze most parameters of the pretrained Mask2Former, fine-tuning only its classification and mask head, as well as the proposed flow module. We apply loss functions in Mask2Former, including a classification loss and a binary mask loss on the class embeddings and warped mask embeddings, respectively. All the models are trained with the AdamW optimizer [41] for a maximum of 90k iterations and the polynomial learning rate decay schedule [4] with an initial learning rate of 5e-5.

**Compared methods.** To validate the effectiveness of our proposed model, we include the following methods for study: *Per-frame*: Employing Mask2Former [7] to independently predict masks for each video frame. *Optical flow*: Warping mask predictions from key frames to non-key frames using optical flow estimated by a lightweight optical flow model, *i.e.*, FlowNet [12]. *Query-random*: Implementing the proposed flow estimation method with randomly initialized flow queries. *Query-learned*: Utilizing the proposed flow estimation method using the learned queries $\mathcal{Q}_O^k$ as initialization. *Query-for-PF*: Using pixel-wise flow $\mathcal{F}^{PF}$ which incorporates segment-level information from the learned flow queries for warping predictions.

**Evaluation metrics.** Following previous works, we use mean Intersection over Union (mIoU) at single-scale inference, and Weighted IoU (WIoU) to evaluate the segmentation performance. We also compare models in terms of their computational complexity with FLOPs to evaluate the efficiency of these VSS models. We calculate FLOPs for each single frame by averaging on a video clip with 15 frames and fix the image resolution of $480 \times 853$ and $1024 \times 2048$ for VSPW and Cityscapes, respectively. For VSPW dataset, we adopt video consistency (VC) [42] to evaluate the visual smoothness of the predicted segmentation maps across the temporal dimension.

Table 2: Performance comparisons with the state-of-the-art VSS methods on Cityscapes. We report mIoU for the semantic segmentation performance and FLOPs (G) for the computational cost comparison, which is averaged by a video clip of 15 images with a resolution of $1024 \times 2048$. Frame-per-second (FPS) is measured on a single NVIDIA V100 GPU with 3 repeated runs.

| Methods | Backbone | mIoU | GFLOPs | Params(M) | FPS |
|---------|----------|------|--------|-----------|-----|
| FCN [40] | R101 | 76.6 | 2203.3 | 68.5 | 2.83 |
| PSPNet [75] | R101 | 78.5 | 2048.9 | 67.9 | 2.88 |
| DFF [82] | R101 | 68.7 | 100.8 | - | - |
| DVSN [66] | R101 | 70.3 | 978.4 | - | - |
| Accel [22] | R101 | 72.1 | 824.4 | - | - |
| ETC [38] | R18 | 71.1 | 434.1 | - | - |
| SegFormer [62] | MiT-B1 | 78.5 | 243.7 | 13.8 | 20.7 |
| SegFormer | MiT-B5 | 82.4 | 1460.4 | 84.7 | 7.20 |
| CFFM-VSS [53] | MiT-B0 | 74.0 | 80.7 | 4.6 | 15.79 |
| CFFM-VSS | MiT-B1 | 75.1 | 158.7 | 15.4 | 11.71 |
| MRCFA [54] | MiT-B0 | 72.8 | 77.5 | 4.2 | 16.55 |
| MRCFA | MiT-B1 | 75.1 | 145 | 14.9 | 12.97 |
| Mask2Former [7] | R50 | 79.4 | 529.9 | 44.0 | 6.58 |
| Mask2Former | R101 | 80.1 | 685.5 | 63.0 | 5.68 |
| Mask2Former | Swin-T | 82.1 | 543.6 | 47.4 | 5.41 |
| Mask2Former | Swin-S | 82.6 | 730.1 | 68.7 | 4.31 |
| Mask2Former | Swin-B | 83.3 | 1057.0 | 107.0 | 3.26 |
| Mask2Former | Swin-L | 83.3 | 1911.3 | 215.0 | 2.11 |
| Mask2Former-DFF | R101 | 77.1 | 457.4 | 101.7 | 7.14 |
| Mask2Former-DFF | Swin-B | 79.9 | 525.3 | 145.7 | 6.09 |
| Mask2Former-Accel | R101+R50 | 78.9 | 594.8 | 145.7 | 5.78 |
| Mask2Former-Accel | Swin-B+Swin-T | 81.4 | 680.1 | 193.1 | 4.40 |
| MPVSS | R50 | 78.4 | 173.2 | 84.1 | 13.43 |
| MPVSS | R101 | 78.2 | 204.3 | 103.1 | 12.55 |
| MPVSS | Swin-T | 80.7 | 175.9 | 114.0 | 12.33 |
| MPVSS | Swin-S | 81.3 | 213.2 | 108.0 | 10.98 |
| MPVSS | Swin-B | 81.7 | 278.6 | 147.0 | 9.54 |
| MPVSS | Swin-L | 81.6 | 449.5 | 255.4 | 7.24 |

## 4.2 Comparison with State-of-the-art Methods

The comparisons on VSPW dataset are shown in Tab. 1. We also report the backbone used by each method and the computational complexity (GFOLPs), averaged over a video clip of 15 frames, with each frame of resolution 480×853. For our proposed models, we use 5 as the default key frame interval for comparison. We have the following analysis: **1)** The proposed MPVSS achieves comparable accuracy with significantly reduced computational cost when compared to the strong baseline method, Mask2Former [7]. Specifically, compared to Mask2Former, our MPVSS reduces the computational cost in terms of FLOPs by 71.7G, 96.2G, 74.7G, 104.9G, 162.0G, and 305.4G on R50, R101, Swin-T, Swin-S, Swin-B, and Swin-L backbones, respectively, while only experiencing 1.0%, 0.5%, 1.3%, 1.7%, 1.5%, and 2.2% degradation in the mIoU score. These results demonstrate a promising trade-off between accuracy and efficiency of our framework. **2)** Our models with different backbones perform on par with the state-of-the-art VSS methods on the VSPW dataset with much fewer FLOPs. Specifically, our MPVSS model with the Swin-L backbone achieves an impressive mIoU of 53.9%, surpassing CFFM-VSS with MiT-B5 backbone by 4.6%. Notably, our approach achieves this performance with only 24% of the computational cost in terms of FLOPs. **3)** When considering the comparable computational cost, our model consistently outperforms other state-of-the-art methods. For instance, MPVSS with Swin-L backbone surpasses Segformer with MiT-B2 backbone by 10% mIoU, under the FLOPs around 3.5G. These compelling results highlight the exceptional performance and efficiency of our MPVSS against the compared VSS approaches. In terms of temporal consistency, our proposed MPVSS achieves comparable mVC scores when compared to approaches that explicitly exploit cross-frame temporal context [54] or incorporate temporal consistency constraints during training [38]. We provide an explanation of VC score and a discussion in the supplementary material.

We further evaluate our MPVSS on the Cityscapes dataset and achieve state-of-the-art results with relatively low computational complexity. Notably, the Cityscapes dataset only provides sparse anno-

Table 3: Ablation study on VSPW dataset.

(a) Effect of the proposed flow estimation method.

| Backbone | Per-frame | Copy | Optical Flow | Ours |
|---|---|---|---|---|
| R50 | 38.5 | 35.7 | 36.1 | **37.5** |
| R101 | 39.3 | 37.1 | 38.2 | **38.8** |
| Swin-T | 41.2 | 37.5 | 38.4 | **39.9** |
| Swin-S | 42.1 | 38.7 | 40.1 | **40.4** |
| Swin-B | 54.1 | 49.3 | 52.1 | **52.9** |
| Swin-L | 56.1 | 50.4 | 53.2 | **53.9** |

(b) Effect of each component.

| Flow design | Params(M) | Swin-T | Swin-B |
|---|---|---|---|
| Optical Flow | 38.68 | 38.4 | 52.1 |
| Query-random | 37.56 | 38.1 | 50.9 |
| Query-learned | 37.56 | 39.2 | 52.4 |
| Query-for-PF | 40.08 | 38.5 | 52.3 |
| Query-based flow maps | 40.08 | 39.9 | 52.9 |

tations for one out of every 30 frames in each video. Therefore, for a fair comparison with previous efficient VSS methods, we report the results following the same evaluation process and compute their average FLOPs per-frame over all video frames. Specifically, compared to Mask2Former, our `MPVSS` reduces the computational cost in terms of FLOPs by 0.36T, 0.48T, 0.37T, 0.52T, 0.78T and 1.5T on R50, R101, Swin-T, Swin-S, Swin-B, and Swin-L backbones, respectively, with only 1.0%, 1.9%, 1.4%, 1.3%, 1.6% and 1.7% degradation in the mIoU score. The strong accuracy-efficiency trade-offs demonstrate that our proposed framework also works on sparse annotated video data.

## 4.3 Ablation Study

**Effects of the query-based flow.** We compare the performance between the proposed query-based flow and the traditional optical flow in Table 3a. Overall, our approach demonstrates superior performance compared with the models relying solely on optical flow for mask propagation from key frames to non-key frames. To be specific, the utilization of the proposed query-based flow for mask warping consistently boosts the performance by 1.4%, 0.6%, 1.5%, 0.3%, 0.8% and 0.7% in terms of mIoU on R50, R101, Swin-T, Swin-S, Swin-B and Swin-L, respectively, which leads to less degradation to the per-frame baseline compared to models using pixel-wise optical flow. Moreover, directly copying the mask predictions from the key frame leads to a clear performance drop.

**Effects of each component.** In Table 3, we verify the effects of each component design in our query-based flow. We conduct experiments on VSPW using Swin-T and Swin-B. The performance of *Query-random* on the two backbones is slightly inferior to using optical flow. This suggests that without per-segment query initialization, the query-based flow struggles to retrieve specific motion information from motion features for one-to-one mask warping from the key frame, resulting in suboptimal performance. Then, by using key-frame queries as the initialization for flow queries, we observe an improvement of 1.1% and 1.5% mIoU scores for *Query-learned* on Swin-T and Swin-B respectively, compared to *Query-random*. Furthermore, we develop a variant design *Query-for-PF*, which introduces additional performance gain by 0.1% and 0.2% compared to optical flow respectively. These results underscore two findings: **1)** learning query-based flow for each segment requires the learned queries from the key frame, for extracting meaningful motion features for each segment; **2)** integrating segment-level information benefits the learning of pixel-wise flow. Therefore, we incorporate a refinement step using the enhanced pixel-wise flow for producing each query-based flow, which results in the best performance on both backbones. The outcomes highlight that warping mask predictions using our query-based flow consistently outperforms the methods using traditional optical flow on VSS.

**Effects of the key frame interval.** In Fig. 3a and Fig. 3b, we show the mIoU scores and TFLOPs versus key frame interval for our `MPVSS` with different backbones, respectively. Overall, all the results achieve significant FLOPs reduction with decent accuracy drop, which smoothly trade in accuracy for computational resource and fit different application needs flexibly. In Fig. 3c, we present the performance of warping mask predictions using our query-based flow and traditional optical flow, as well as direct copying version, on Swin-T. We observe that, when the key frame interval increases from 2 to 10, the performance of the proposed query-based flow only slightly degraded by 1.1% and 0.7% in terms of mIoU and WIoU score, while the scores are decreased by 2.63% and 2.96% using optical flow. These results indicates that the proposed query-based flow is more robust to capture long-term temporal changing for videos. We provide comparisons on other backbones in the supplementary material.

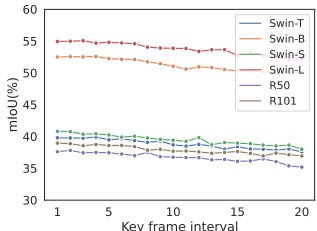
(a) mIoU vs. Key frame interval with different backbones.

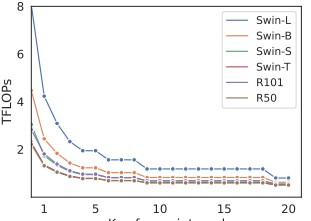
(b) TFLOPs vs. Key frame interval with different backbones.

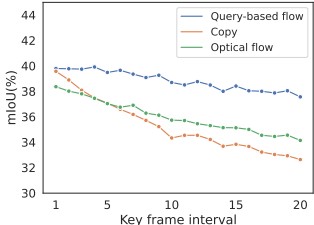
(c) mIoU vs. Key frame interval of different warping method.

Figure 3: Effects of the key frame interval. We compare the trade-off between accuracy and efficiency on variant backbones and warping strategies with respect to key frame interval. The number of FLOPs (T) is calculated based on a clip of 15 frames with a resolution of $480 \times 853$.

## 4.4 Qualitative Results

In Fig. 4, we present two examples from VSPW dataset to visually compare the prediction quality of the per-frame baseline, the method using optical flow, and the proposed MPVSS. We observe that the proposed query-based flow exhibits superior performance in capturing the movement of small objects when compared to optical flow. Additionally, the proposed mask propagation framework demonstrates enhanced category consistency compared to the per-frame baseline. For instance, the per-frame baseline exhibits category inconsistency as each frame is independently predicted. This highlights the advantage of our proposed approach in maintaining category coherence across consecutive frames.

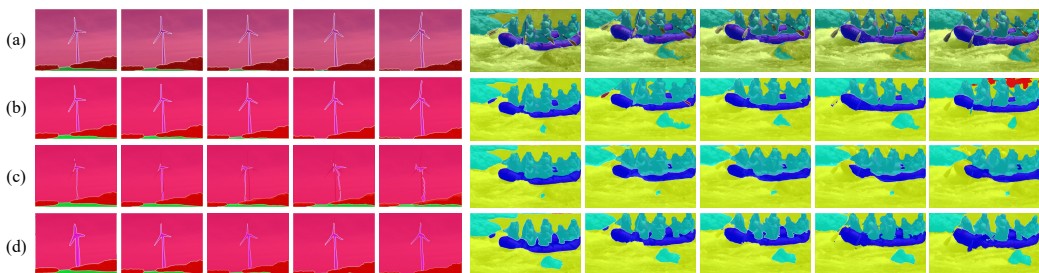

Figure 4: Qualitative results. We present two examples from VSPW dataset. In each example, we display a series of consecutive frames from left to right. From top to bottom: (a) the ground truth; (b) the predictions of per-frame baseline, (c) the predictions of method using optical flow; (d) the predictions of the proposed MPVSS.

## 5 Conclusion and Future Work

In this paper, we have presented a simple yet effective mask propagation framework, dubbed MPVSS, for efficient VSS. Specifically, we have employed a strong query-based image segmentor to process key frames and generate accurate binary masks and class predictions. Then we have proposed to estimate specific flow maps for each segment-level mask prediction of the key frame. Finally, the mask predictions from key frames were subsequently warped to other non-key frames via the proposed query-based flow maps. Extensive experiments on VSPW and Cityscapes have demonstrated that our MPVSS achieves SOTA accuracy and efficiency trade-off. Future work could explore extending MPVSS to other video dense prediction tasks, *e.g.*, video object and instance segmentation.

**Limitations and broader impacts.** Although our framework significantly reduces computational costs for processing videos, the new module leads to an increase in the number of parameters in the models. Additionally, there might be a need for fine-tuning the new module or optimizing hyperparameters to achieve the best performance. It can lead to increased carbon emissions, indicating the potential negative societal impact.

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
