# Supplementary Material for Mask Propagation for Efficient Video Semantic Segmentation

**Yuetian Weng**[1,2]* **Mingfei Han**[3,4] **Haoyu He**[1] **Mingjie Li**[3]
**Lina Yao**[4] **Xiaojun Chang**[3,5] **Bohan Zhuang**[1]†
[1]ZIP Lab, Monash University    [2]Baidu Inc.    [3]ReLER, AAII, UTS
[4]Data61, CSIRO    [5]Mohamed bin Zayed University of AI

We organize our supplementary material as follows:

- In Section A, we present more analytical results on VSPW dataset.

- In Section B, we provide more ablation studies on Cityscapes dataset.

- In Section C, we present qualitative results on Cityscapes dataset.

- In Section D, we provide computational cost analysis and training details.

- In Section E, we provide comparison with bi-directional optical flow.

## A    More Analysis on VSPW

### A.1    Explanation of Video Consistency

Following [3], we use Video Consistency (VC) to evaluate the category consistency among adjacent frames in the videos. $\text{VC}_f$ for $f$ consecutive frames in a video clip $\{\mathbf{I}^t\}_{t=1}^{T}$ is computed by $\text{VC}_f = \frac{1}{T-f+1}\Sigma_{i=1}^{T-f+1} \frac{(\cap_i^{i+f-1}\mathbf{P}^i)\cap(\cap_i^{i+f-1}\hat{\mathbf{P}}^i)}{\cap_i^{i+f-1}\mathbf{P}^i}$, where $T \geq f$. $\mathbf{P}^i$ and $\hat{\mathbf{P}}^i$ are the ground-truth mask and predicted mask for the $i^{th}$ frame, respectively. We compute the mean of $\text{VC}_f$ for all videos in the datasets and report $\text{mVC}_8$, $\text{mVC}_{16}$ following [3].

Table 1 in Section 4 displays the comparisons of mVC scores on VSPW dataset. Notably, our proposed MPVSS achieves comparable mVC scores when compared to approaches that explicitly exploit cross-frame temporal context [4] or incorporate temporal consistency constraints during training [2]. This observation demonstrates that our mask propagation framework implicitly captures the long-range temporal relationships among video frames. Furthermore, the proposed mask propagation framework outperforms the per-frame baseline on both $\text{mVC}_8$ and $\text{mVC}_{16}$, indicating its effectiveness in mitigating the inconsistency issues observed in the predictions of the per-frame baseline through the process of mask propagation.

### A.2    Effects of the Number of Stages in Motion Decoder

In Table I, we ablate the effects of the number of decoder stages $S$ in the motion decoder module, conducted on VSPW dataset. On both backbones of Swin-T and Swin-B, the mIoU scores first increase as the number of decoder stages increases to 3 and then drop at the number of 4. Finally, we adopt the number of decoder stages $S = 3$ in our model.

---

*Work done during an internship at Baidu Inc.
†Corresponding author. Email: bohan.zhuang@gmail.com

37th Conference on Neural Information Processing Systems (NeurIPS 2023).

Table I: Effects of the number of stages in motion decoder.

| # of decoder stages | Swin-T | Swin-B |
|:---:|:---:|:---:|
| 1 | 39.4 | 51.8 |
| 2 | 39.4 | 52.0 |
| 3 | 39.9 | 52.6 |
| 4 | 39.6 | 51.8 |

## A.3   Effects of the Key Frame Interval

Figure A compares the mIoU scores versus key frame interval on diverse backbones for (1) *Query-based flow*: the proposed MPVSS, which employs query-based flow for mask propagation; (2) *Optical flow*: mask warping by using traditional optical flow; (3) *Copy*: directly copying mask predictions from key frames to the following non-key frames. In particular, directly copying predictions from the key frame to non-key frames results in a noticeable drop in performance as the key frame interval increases. When the key frame interval ranges from 2 to 10, the performance of MPVSS utilizing the proposed query-based flow exhibits only a slight degradation of 0.9%, 1.3%, 1.4%, 1.5% on R50, R101, Swin-T, Swin-S and Swin-B backbones, respectively. However, on Swin-L backbone, the performance reduction amounts to 2.7%. Furthermore, our observations indicate that the proposed MPVSS outperforms the optical flow counterpart on R50, R101, and Swin-T backbones as the key frame interval increases.

## A.4   Per-category Analysis

In Figure B, we present the per-category mIoU scores of the per-frame baseline, the proposed query-based flow (mask propagation) and optical flow, using Swin-L as the backbone, on VSPW dataset. Figure B (a) highlights the categories where the proposed query-based flow outperforms traditional optical flow, while Figure B (b) showcases some worse-performing cases. Among the 124 categories in VSPW dataset, the proposed query-based flow outperforms optical flow in 91 categories. Remarkably, the query-based flow demonstrates exceptional performance in outdoor scenes and dynamic categories like "Train" and "Truck". However, there are specific categories, such as "Sea" and "Water", where the proposed method exhibits relatively lower performance. This difference in performance could be due to challenges in correctly classifying similar concepts, such as distinguishing between "Sea", "Water" or "Lake".

# B   More Results on Cityscapes

Table II: Effects of the query-based flow on Cityscapes dataset.

| Backbone | Per-frame | Optical flow | Query-learned | Query-based flow |
|:---:|:---:|:---:|:---:|:---:|
| R50 | 79.4 | 76.6 | 77.3 | 78.4 |
| R101 | 80.1 | 77.7 | 77.8 | 78.2 |
| Swin-T | 82.1 | 79.6 | 80.2 | 80.7 |
| Swin-S | 82.6 | 80.4 | 80.7 | 81.3 |
| Swin-B | 83.3 | 80.5 | 80.8 | 81.7 |
| Swin-L | 83.3 | 80.7 | 81.2 | 81.6 |

Table II presents more results on Cityscapes dataset. We compare (1) *Per-frame*: Using Mask2Former [1] to predict masks on each frame independently; (2) *Optical flow*: Warping mask predictions using optical flow estimated by FlowNet; (3) *Query-learned*: Mask propagation using the proposed flow map estimation with learned queries from key frames as initialization but without using pixel-wise flow for refining; (4) *Query-based flow*: Mask propagation using the proposed query-based flow. *Query-learnd* achieves better performance in terms of mIoU scores compared to *Optical flow*. While our MPVSS that utilizes the proposed *Query-based flow* leads to consistent improvements of

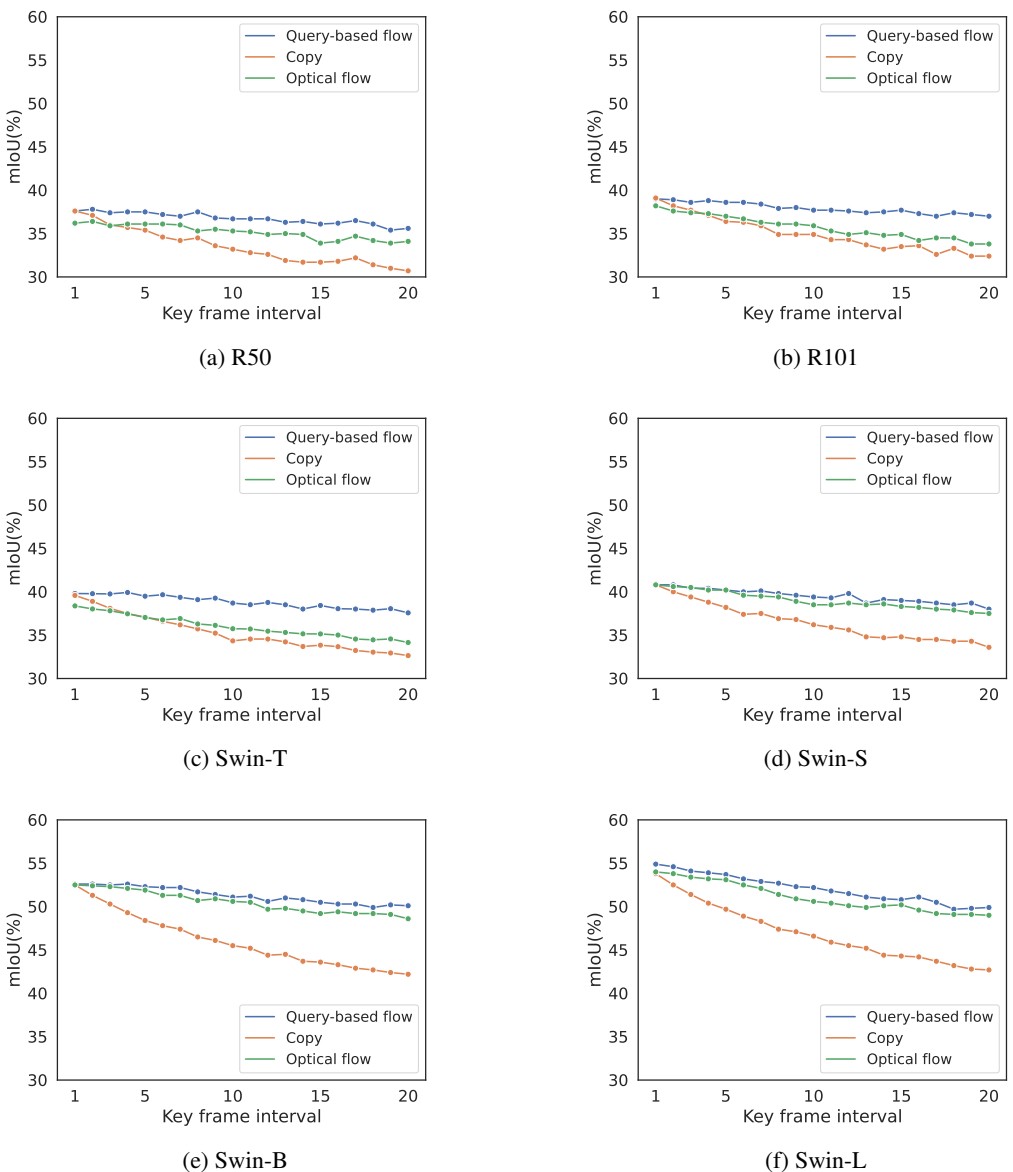

Figure A: mIoU vs. Key frame interval of different warping methods on diverse backbones.

1.8%, 0.5%, 1.1%, 0.9%, 1.2% and 0.9% in terms of mIoU on R50, R101, Swin-T, Swin-S, Swin-B and Swin-L, respectively.

## C Qualitative Results

We provide visualization of the query-based flow and pixel-wise flow in Figure C (a)-(c). The query-based flow and pixel-wise flow are compensated. The query-based flow captures the overall motion for the specific segment, while the pixel-wise flow adds more detailed movement at the fine-grained level. Fusing the two kinds of flow results in more stable flow maps for mask propagation.

Furthermore, we provide a visual comparison between utilizing the learned queries $\mathcal{Q}_{\mathcal{O}}^k$ through the flow maps and the attention maps derived from the motion decoder. As shown in row (f)(g) and row (c)(h) in Figure C, with $\mathcal{Q}_{\mathcal{O}}^k$ initialization, the associated attention maps focus on particular regions of the motion map, leading to improved flow maps for the corresponding segments.

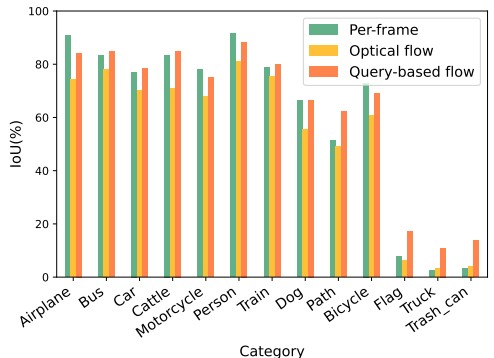
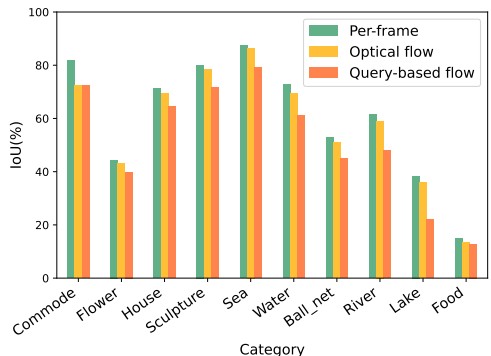

(a) Better case: query-based flow outperforms on numerous dynamic categories.

(b) Worse case: query-based flow exhibits relatively lower performance on some categories.

Figure B: Per-category analysis.

## D    Computational Costs and Training Time

We provide detailed computational costs for a single key frame and a non-key frame of the proposed MPVSS in Table III. The main computational demands lie in the computation of a high-resolution pixel-embedding. Given that the semantic information in consecutive video frames is highly correlated, we propose to propagate accurate predicted masks from a key-frame to its subsequent non-key frames, achieving a better accuracy and efficiency trade-off.

Table III: Computational cost in terms of FLOPs (G) for a single key frame and a non-key frame that using different backbones.

| Dataset | Backbone | R50 | R101 | Swin-T | Swin-S | Swin-B | Swin-L |
|---------|----------|-----|------|--------|--------|--------|--------|
| VSPW | Key-frame
Non-key frame | 110.6 | 141.3 | 114.4 | 152.2
21.0 | 223.5 | 402.7 |
| Cityscapes | Key-frame
Non-key frame | 529.9 | 685.5 | 543.6 | 730.1
84.0 | 1057.0 | 1911.3 |

We present the training hours in Table IV. During training, we sample a pair of frames, *i.e.*, a key frame and a non-key frame. The segmentation network is applied to the key frame, and a set of query-based flow maps is estimated by the proposed flow module. These flow maps are then used to obtain the warped mask embedding for the non-key frame. We apply loss functions in Mask2Former, including a classification loss and a binary mask loss on the class embeddings and warped mask embeddings, respectively. Bipartite matching is performed between the warped mask embeddings and the ground truth masks. Subsequently, the loss gradients are propagated backward across the model to update the proposed flow module.

Table IV: Training time (Hours) on VSPW dataset. Experiments are conducted with a batch size of 16 on 8 NVIDIA V100 GPUs.

| Backbone | R50 | R101 | Swin-T | Swin-S | Swin-B | Swin-L |
|----------|-----|------|--------|--------|--------|--------|
| Mask2Former | 11.4 | 12.3 | 10.3 | 12.0 | 12.3 | 15.0 |
| MPVSS | 9.3 | 11.5 | 10.2 | 11.1 | 12.2 | 13.4 |

## E    Comparison with Bi-directional Optical Flow

In the task of Video Object Segmentation (VOS), propagating masks from previous key frames often leads to occlusion issues on the predicted masks, causing noticeable ghosting effects. [5] proposes

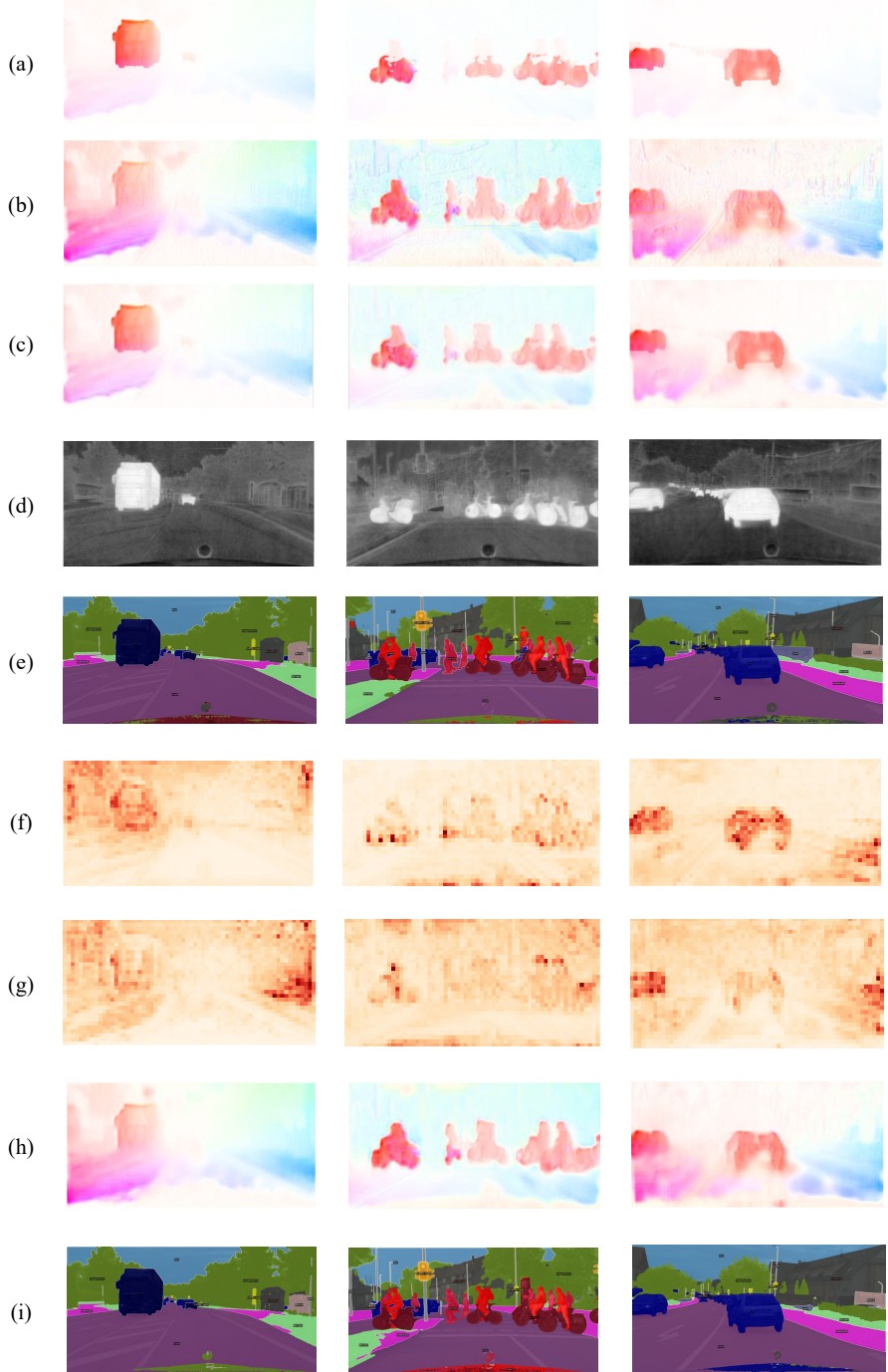

Figure C: Qualitative results. From left to right: three samples from Cityscapes dataset. From top to bottom: (a) Query-based flow corresponding to flow queries; (b) Pixel-wise flow; (c) Flow maps for the corresponding queries; (d) Warped mask predictions; (e) Semantic predictions; (f) Attention maps of the flow queries (with $\mathcal{Q}_{\mathcal{O}}^k$ initialization); (g)-(i) Attention maps, flow maps for the model of flow query without $\mathcal{Q}_{\mathcal{O}}^k$ initialization and the predicted semantic maps.

Table V: Accuracy comparison (mIoU) for uni-directional optical flow, bi-directional optical flow and the proposed query-based flow. We also report computational efficiency in terms of FLOPs for each single non-key frame.

| Backbone | Uni-directional OF | Bi-directional OF | Query-based flow |
|---|---|---|---|
| Swin-T | 79.6 | 80.2 | 80.7 |
| Swin-B | 80.5 | 81.1 | 81.7 |
| FLOPs (G) | 47.3 | 89.6 | 84.7 |

to integrate bi-directional motion vector encoded within compressed videos for mask propagation. The occurrence of ghosting effects appears to be less pronounced when our network is applied to VSS datasets in comparison to VOS. One potential explanation could be the smoother transitions between scenes commonly observed in VSS datasets. In contrast, the ghosting effects encountered in VOS datasets often stem from delayed responses to rapidly moving objects. Moreover, our query-based flow, which integrates segment-level information, might provide more stable and robust mask propagation. For further investigation between uni-directional optical flow, query-based flow, and bi-directional optical flow, we conduct experiments on Cityscapes and report results in Table V.

**Query-based flow vs. uni-directional optical flow.** As discussed in Section 4 in the main paper, the proposed query-based flow is more robust to capture long-term temporal changes compared to using unidirectional optical flow.

**Bi-directional optical flow vs. uni-directional optical flow.** As shown in Table V, bi-directional optical flow outperforms uni-directional optical flow in terms of mIoU scores as it fuses accurate semantic maps from two key frames.

**Query-based flow vs. bi-directional optical flow.** The proposed query-based flow achieves slightly better mIoU scores compared to bi-directional flow with fewer FLOPs for every single non-key frame. To delve deeper into the segmentation performance, we visualized the qualitative results in Figure D. Bi-directional optical flow proves ineffective when the two warped masks are not aligned well due to inaccurate pixel-wise flow prediction. Nevertheless, the proposed query-based flow provides clear boundaries and compensations for warping masks of irregular or small semantic segments.

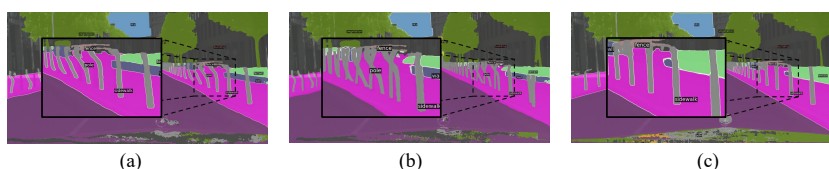

(a)          (b)          (c)

Figure D: Visualization. We provide the semantic maps of model using (a) uni-directional flow; (b) bi-directional optical flow and (c) our query-based flow.