# OpenReview forum: "Mask Propagation for Efficient Video Semantic Segmentation"
_NeurIPS.cc/2023/Conference — NeurIPS 2023 poster_

### Official Review · Reviewer_cyWY · 2023-07-04

**Soundness:** 3 good
**Presentation:** 3 good
**Contribution:** 3 good
**Rating:** 6
**Confidence:** 4

**Summary:**

This paper presents a mask propagation method, MPVSS, for video semantic segmentation (VSS). MPVSS uses Mask2Former to obtain mask predictions and queries from the key frame. Then, a motion encoder extracts pixel-wise motion from the key frame and its adjacent frame. The queries from the key frame are used to decode query-wise motion features. A flow head generates query-based flow maps, and binary mask predictions for the adjacent frame are generated by applying these flow maps to the binary mask predictions of the key frame. Semantic segmentations of non-key frames are produced by matrix multiplication between the key frame's query classification output and the binary mask prediction for the adjacent frame. A detailed ablation study validates each component. Comparisons between MPVSS and state-of-the-art methods on VSPW and Cityscapes demonstrate good performance and efficiency of MPVSS.

**Strengths:**

This paper is well-written and easy to follow.
Motivation leveraging motion estimation to reduce the computation of VSS is sound.
The proposed method of propagating the key frame’s query to the motion features is interesting and novel.
The experiments are well-designed, and the results are convincing.



**Weaknesses:**

The key frames are selected at fixed intervals. As a result, the method may not fully address the mentioned redundancy problem.

Efficiency of the proposed method should be discussed in more detail. For instance, MPVSS uses FlowNet to extract motion features between two frames instead of relying on a backbone network to extract frame-wise features.

Fig3 (b), y axis label, “TLOPs” seems a typo.

There is not much information about training time, and any particular training strategies used in the experiments.



**Questions:**

Please see Weaknesses

**Limitations:**

The paper adequately addresses limitations.

---

> ### Author Rebuttal · Authors · 2023-08-10
>
>
> Thanks for your constructive comments and we address your questions as follows.
>
> **Q1.** The method may not fully address the mentioned redundancy problem as the key frames are selected at fixed intervals.
>
> **A1.** First, we acknowledge that employing fixed key-frame intervals only results in a partial reduction of redundancy, which is a limitation of the current mask propagation framework. Secondly, a more effective approach would involve utilizing dynamic key-frame selection. However, this may introduce additional model parameters and potentially elevate computational costs due to the decision-making process for key-frame selection. In the future, we will consider exploring the implementation of dynamic key-frame selection to address the temporal redundancy challenge more comprehensively.
>
> **Q2.** Efficiency of the proposed method should be discussed in more detail.
>
> **A2.** We provide detailed computational costs for a single key frame and a non-key frame of the proposed MPVSS in Table E. As mentioned in Lines 43-44, the main computational demands lie in the computation of a high-resolution pixel-embedding. Utilizing the strong semantic correlation in consecutive video frames, we propose to propagate accurately predicted masks from a key-frame to its subsequent non-key frames. It only takes the cost of the flow module and mask propagation process, leading to the accuracy-efficiency trade-off.
>
> Table E. Computational cost in terms of FLOPs (G) for a single key frame and a non-key frame that uses different backbones. We use input resolutions of 480x853 and 1024x2048 for VSPW and Cityscapes, respectively.
>
> | Dataset    |               | R50   | R101  | Swin-T | Swin-S | Swin-B | Swin-L |
> | ---------- | ------------- | ----- | ----- | ------ | ------ | ------ | ------ |
> | VSPW       | Key-frame     | 110.6 | 141.3 | 114.4  | 152.2  | 223.5  | 402.7  |
> |  | Non-key frame | 21.0 |  21.0     |   21.0     |  21.0      |21.0|21.0|
> | Cityscapes | Key-frame     | 529.9 | 685.5 | 543.6  | 730.1  | 1057.0 | 1911.3 |
> |  | Non-key frame | 84.0 |84.0|84.0|84.0|84.0|84.0|
>
> **Q3.** Training time and training strategies.
>
> **A3.** We list the training hours in Table F. The training strategies for the proposed MPVSS have been outlined in Lines 168-171 and Lines 263-264. During training, we sample a pair of frames, i.e., a key frame and a non-key frame. We run the segmentation network on the key frame, and a set of query-based flow maps are estimated by the proposed flow module, which are then used to obtain the warped mask embedding for the non-key frame. We apply loss functions in Mask2Former, including a classification loss and a binary mask loss on the class embeddings and warped mask embeddings, respectively. Bipartite matching is performed between the warped mask embeddings and the ground truth masks. Subsequently, the loss gradients are propagated backward across the model to update the proposed flow module.
>
> Table F. Training time (Hours) on VSPW dataset. Experiments are conducted with a batch size of 16 on 8 NVIDIA V100 GPUs.
>
> | Backbone    | R50  | R101 | Swin-T | Swin-S | Swin-B | Swin-L |
> | ----------- | ---- | ---- | ------ | ------ | ------ | ------ |
> | Mask2Former | 11.4 | 12.3 | 10.3   | 12.0   | 12.3   | 15.0   |
> | MPVSS       | 9.3  | 11.5 | 10.2   | 11.1   | 12.2   | 13.4   |

---

> > ### Comment · Reviewer_cyWY · 2023-08-16
> >
> > I have read the rebuttal and remain with my initial rating.

---

### Official Review · Reviewer_LSXE · 2023-07-04

**Soundness:** 3 good
**Presentation:** 3 good
**Contribution:** 3 good
**Rating:** 5
**Confidence:** 4

**Summary:**

An efficient mask propagation framework for VSS, called MPVSS, is proposed in this paper.

MPVSS adopts a strong query-based image segmentation based on sparse key frames and warp prediction to non-key frames by generating a segment-aware flow from a newly designed flow estimation module.

With the proposed query-based flow, MPVSS achieves the performance of SOTA with a better efficiency.

**Strengths:**

1. MPSVSS captures segment-level information between the video frames, which provides a better propagation result than the pixel-level correspondence information used in the previous work.

2. Compared with vanilla optical flow, query-based flow provides smooth and clear compensation for non-keyframes.

3. The proposed method achieves SOTA performance with significantly reduced computational cost through extensive experiments on standard benchmarks.

4. MPVSS delivers a consistently higher level of video consistency than its base network.

**Weaknesses:**

1. typo: line 10: wrapped -> warped.

2. The proposed mask propagation framework is identical to that of DFF [1], Accel [2], and CoVOS [3], while the proposed query-based flow requires further experimentation to demonstrate its effectiveness. More specifically:
- In Table 2, it is difficult to say whether the query-based flow is better than Accel [2] or DFF [1], because the MPVSS uses Mask2Former for key-frames segmentation. In order to prove the advantage of propagation using query-based flow over the propagation module in Accel and DFF, the keyframe segmentation network should be unified.
- In Table 3a, the query-based flow does bring improvement, but it also introduces additional network parameters. It's hard to say whether the improvement is due to the increase in parameters.
3. Since propagation relies only on the previous keyframe, I wonder how occlusion will have a negative effect on the propagation results. There is literature using bi-directional motion vector for mask propagation [3] and shows that occlusion has a strong effect on the propagation results, especially causing ghosting effect, however this point is not discussed in the paper.
4. A measurement on the FPS should be provided.

[1] Zhu, Xizhou, et al. "Deep feature flow for video recognition." CVPR. 2017.

[2] Jain, Samvit, Xin Wang, and Joseph E. Gonzalez. "Accel: A corrective fusion network for efficient semantic segmentation on video." CVPR. 2019.

[3] Xu, Kai, and Angela Yao. "Accelerating video object segmentation with compressed video." CVPR. 2022.


**Questions:**

Please refer to the Weaknesses section.

**Limitations:**

The authors adequately addressed limitation.

---

> ### Author Rebuttal · Authors · 2023-08-10
>
>
> We thank you for your valuable feedback and address your questions as follows.
>
> **Q1.** Fair comparison between DFF and Accel by unifying the key frame segmentation network.
>
> **A1.** Thanks for your advice. We integrate DFF[78] and Accel[20] with Mask2Former to conduct fair comparisons on Cityscapes dataset. Compared with Mask2Former-DFF, we achieve higher mIoU scores and lower GFLOPs. And the proposed MPVSS surpasses Mask2Former-Accel on Swin-B by 0.3% with only 1/3 GFLOPs.
>
>
> Table C. Performance comparisons between DFF and Accel.
>
> |      Methods      |   Backbone    | mIoU | GFLOPs | #Params (M) |  FPS  |
> | :---------------: | :-----------: | :--: | :----: | :---------: | :---: |
> |  Mask2Former-DFF  |     R101      | 77.1 | 457.4  |    101.7    | 7.14  |
> |  Mask2Former-DFF  |    Swin-B     | 79.9 | 525.3  |    145.7    | 6.09  |
> | Mask2Former-Accel |   R101+R50    | 78.9 | 594.8  |    145.7    | 5.78  |
> | Mask2Former-Accel | Swin-B+Swin-T | 81.4 | 680.1  |    193.1    | 4.40  |
> |       MPVSS       |     R101      | 78.2 | 204.3  |    103.1    | 12.55 |
> |       MPVSS       |    Swin-B     | 81.7 | 278.6  |    147.0    | 9.54  |
>
>
> **Q2.** It's hard to say whether the improvement is due to the increase in parameters.
>
> **A2.** Compared with methods that warp masks by optical flow, our *Query-based* flow introduces additional ~1.4M parameters, which can be considered negligible. Additionally, in Table D, we measure the number of parameters for the flow module listed in Table 3(b) of the main paper. Using *Query-learned flow* already achieves better performance than optical flow with a lower number of parameters, which also proves the improvement mainly comes from the effectiveness of the module design.
>
> Table D. Performance comparisons for different flow designs. We report the number of parameters of different flow modules, and mIoU scores.
>
> |   Flow design    | #Params (M) | Swin-T | Swin-B |
> | :--------------: | :---------: | :----: | :----: |
> |   Optical flow   |    38.68    |  38.4  |  52.1  |
> |   Query-random   |    37.56    |  38.1  |  50.9  |
> |  Query-learned   |    37.56    |  39.2  |  52.4  |
> |   Query-for-OF   |    40.08    |  38.5  |  52.3  |
> | Query-based flow |    40.08    |  39.9  |  52.9  |
>
>
>
> **Q3.**  I wonder how occlusion will have a negative effect on the propagation results, especially the ghosting effect on VSS.
>
> **A3.** Thanks for your valuable feedback. Indeed, we have observed instances of failure due to occlusion challenges when applying our proposed mask propagation framework to VSS datasets. For example, scenarios in which a car moves behind a tree can lead to inaccuracies in mask propagation, primarily due to the reliance on mask predictions from preceding key frames.
>
> Nevertheless, the occurrence of ghosting effects seems to be less pronounced when our network is applied to VSS datasets in comparison to VOS. One potential explanation for this could be the smoother transitions between scenes that are commonly observed in VSS datasets. In contrast, the ghosting effects encountered in VOS datasets often stem from delayed responses to rapidly moving objects. Additionally, our query-based flow, which integrates segment-level information, might offer more stable and robust mask propagation compared to using uni-directional optical flow.
>
> Moreover, we acknowledge that the integration of bi-directional motion vectors encoded within compressed videos in method [A] currently exceeds the scope of our work. Implementing bi-directional mask propagation appears promising, even though it might introduce additional model parameters and computational costs. This is deserving of further exploration, and we regard it as a potential area for future research. We will add a comprehensive discussion of the occlusion challenges in the revised version of our work.
>
>
> **Q4.** Information on FPS should be provided.
>
> **A4.** We measure the FPS of different methods on a single NVIDIA V100 GPU. The results are provided in Table A and Table B in the general response. We will update the corresponding tables in the revised version.
>
> We will fix the typos in the revised version.
>
> **Reference:**
>
> [A] Xu, Kai, and Angela Yao. "Accelerating video object segmentation with compressed video." CVPR. 2022.

---

> > ### Comment · Reviewer_LSXE · 2023-08-15
> >
> > Thanks for the clarifications, I have raised score to 5. I am interested in the fact that you claimed that query-based optical flow can provide more stable and robust mask propagation than unidirectional optical flow. I look forward to the discussion and comparison between unidirectional OF, query-based flow and bidirectional OF.

---

> > > ### Author Response · Authors · 2023-08-18
> > >
> > > Dear Reviewer LSXE,
> > >
> > > Thanks for your feedback and suggestions! We feel glad to address your questions and appreciate the constructive reviews for improving our work.
> > >
> > > For further investigation between uni-directional OF, query-based flow, and bi-directional OF, we run experiments on Cityscapes.
> > >
> > > **Query-based flow vs. uni-directional optical flow.** As discussed in Lines 353-356 in main paper, the proposed query-based flow is more robust to capture long-term temporal changing compared to using unidirectional optical flow.
> > >
> > > **Bi-directional optical flow vs. uni-directional optical flow.** As shown in Table G, bi-directional optical flow outperforms uni-directional optical flow in terms of mIoU scores as it fuses accurate semantic maps from two key frames.
> > >
> > > **Query-based flow vs. bi-directional optical flow.** The proposed query-based flow achieves better mIoU scores compared to bi-directional flow with fewer FLOPs for each single non-key frame. To further look into the segmentation performance, we visualized the qualitative results and found that bi-directional optical flow is ineffective when the two warped masks are not aligned well due to inaccurate *pixel-wise* flow prediction. Nevertheless, the proposed query-based flow provides a clear boundary and compensation for warping masks for irregular or small semantic segments because of the *segment-aware* flow estimation compared to using uni-directional or bi-directional optical flow. Because of the limitations on text-only responses during the discussion phase, we are unable to display visualizations here. We will include more quantitative and qualitative results for better illustration in the revised version.
> > >
> > > Table G: accuracy comparison (mIoU) for uni-directional optical flow, bi-directional optical flow and the proposed query-based flow. We also report computational efficiency in terms of FLOPs for each single non-key frame.
> > >
> > > |   Backbone    | Uni-directional OF | Bi-directional OF | Query-based flow |
> > > | :-----------: | :----------------: | :---------------: | :--------------: |
> > > |    Swin-T     |        79.6        |       80.2        |       80.7       |
> > > |    Swin-B     |        80.5        |       81.1        |       81.7       |
> > > | **FLOPs (G)** |        47.3        |       89.6        |       84.7       |
> > >
> > >
> > >
> > > Best regards,
> > >
> > > Authors of #6943

---

### Official Review · Reviewer_41Ay · 2023-07-06

**Soundness:** 3 good
**Presentation:** 4 excellent
**Contribution:** 3 good
**Rating:** 6
**Confidence:** 4

**Summary:**

This paper presents an approach for video semantic segmentation (VSS) by focusing on the aspect of efficiency. The authors propose a novel mask propagation framework that is built upon a computationally intensive query-based image segmentor called Mask2Former[1]. Instead of processing every frame, the framework leverages the image segmentor to process only the key frames. To generate masks for non-key frames, a novel query-based flow map estimation module is introduced, which predicts optical flows to warp the masks from key frames. The experimental results demonstrate that the proposed framework achieves competitive performance, with only a minimal drop of approximately 1% to 2% when compared to the baseline [1]. Notably, the proposed method achieves these results with significantly reduced Floating Point Operations (FLOPs).

[1] B. Cheng, I. Misra, A. G. Schwing, A. Kirillov, and R. Girdhar. Masked-attention mask transformer for universal image segmentation. In CVPR, pages 1290–1299, 2022.

**Strengths:**

1. The proposed method demonstrates competitive performance on two widely recognized benchmarks, showcasing only a slight decline in the mIoU score, ranging from 1% to 2%. These results are achieved while significantly reducing computation costs (FLOPs).

2. The novel query-based flow estimation module introduced in this paper surpasses traditional pixel-wise optical estimation methods by producing better flow maps. This advancement holds great promise for the field of transformer-based flow estimation.

**Weaknesses:**

Please see my comments in below Questions section.

**Questions:**

1. It would be inappropriate to claim that the proposed method (MPVSS) "achieves SOTA performance". From Table 1 and Table 2, it becomes evident that MPVSS yields slightly lower mIoU scores compared to its baseline counterpart (Mask2Former). Furthermore, Mask2Former already attains SOTA performance when compared to other methods presented in the table, so the primary contribution of MPVSS lies in its efficiency. Please kindly rectify these statements to accurately reflect the contributions of MPVSS.

2. It would be insightful to delve further into the reasons why the learned query from the mask generation branch (Q^k_O) enhances flow estimation. While the paper briefly touches upon this topic in a single sentence (line 220 - line 222), it would be great to include visualizations or attention maps that demonstrate the gain of the learned query on flow estimation.

3. In line 240 - line 242, two flows (query-based flow and pixel-based flow) are stacked to generate the final flow predictions. To gain a better understanding of the effect of the query-based flow, it would be helpful to visualize the query-based flow and pixel-based flow separately. Additionally, it would be beneficial to clarify whether the first row (i.e., Optical flow) in Table 3 (b) refers to the "pixel-based flow".

---

> ### Author Rebuttal · Authors · 2023-08-10
>
> Thank you for the thorough review and constructive questions.
>
> **Q1**. It would be inappropriate to claim that the proposed method (MPVSS) "achieves SOTA performance".
>
> **A1**. Thanks for your advice. We will demonstrate our contribution as "achieve SOTA accuracy and efficiency trade-offs".
>
> **Q2.** Why the learned query from the mask generation branch ($Q^k_O$) enhances flow estimation.
>
> **A2.** (1) Previous studies in DETR-like framework explore the initialization of decoder query and demonstrate that providing good prior of decoder query leads to faster convergence and better performance. (2) Additionally, as mentioned in Lines 220-222, we utilize the learned queries to enable each flow query to extract motion information for each segment appeared in the key frame. (3) We offer a visual comparison between utilizing the learned queries $Q^k_O$ through the flow maps and the attention maps derived from the motion decoder. The visualization results provided in **Figure A** in the general response PDF. As shown in row (f)(g) and row (c)(h) in **Figure A**, with $Q^k_O$ initialization, the associated attention maps focus on particular regions of the motion map, leading to improved flow maps for the corresponding segments.
>
> **Q3.** Visualization of the query-based flow and pixel-wise flow.
>
> We provide the visualization in **Figure A** in the general response PDF. The query-based flow and pixel-wise flow complement each other. The query-based flow captures the overall motion for the specific segment, while the pixel-wise flow adds more detailed movement at a fine-grained level, especially for the edge of the segment. This combination enhances the stability of the final flow maps for mask propagation.
>
> **Q4.** Whether the first row (i.e., Optical flow) in Table 3 (b) refers to the "pixel-based flow".
>
> **A4.** Yes, "Optical flow" in the first row of Table 3(b) refers to the method warping mask predictions using pixel-based flow estimated by an optical flow model.

---

> > ### Comment · Area_Chair_M63S · 2023-08-18
> > **Reviewer 41Ay**
> >
> > Dear Reviewer 41Ay,
> >
> > Could you please read the author's rebuttal and other reviews, and indicate whether your comments have been addressed? Thank you.
> >
> > Best, AC

---

> > > ### Comment · Reviewer_41Ay · 2023-08-20
> > >
> > > I have reviewed the rebuttal and the comments from other reviewers. I am maintaining my original rating - weak accept.
> > >
> > > Additionally, I would recommend incorporating the visualization into the supplementary materials, as it helps to demonstrate the learning of pixel-based flow and query-based flow respectively.

---

### Official Review · Reviewer_8jSp · 2023-07-07

**Soundness:** 2 fair
**Presentation:** 3 good
**Contribution:** 2 fair
**Rating:** 3
**Confidence:** 4

**Summary:**

The paper is to develop an approach to video semantic segmentation via propagating the segmented mask in key frames to non-key frames. Experiments were conducted on several databases with various comparisons.

**Strengths:**

Focusing on improving the computational efficiency for video semantic segmentation;

The paper is well-written.

**Weaknesses:**

It seems that the work lacks of novelty. In my understanding, the key idea is to compute the mask in key frames, and then propagate the segmentation result to non-keyframes, in order to reduce the computation cost. However, in many previous video analysis works, doing many computation in key frames, and then applying the result to non-keyframes, is quite natural.

The segmentation in key frames is performed by using an existing method, Mask2Former [7].

The flow estimation between frames is quite normal: using FlowNet [11] for motion encoding, and transformer based approach [7] for decoding. Not presenting a new method.





**Questions:**

See my comments in the Weakness part.

**Limitations:**

limitation is talked in the paper.

---

> ### Author Rebuttal · Authors · 2023-08-10
>
>
> Thanks for taking the time to review our paper and we address your questions as follows.
>
> **Q1.** Questions about the novelty of the proposed method.
>
> **A1.** As recognized by Reviewers oxr6, 41Ay, and cyWY, the primary innovation of this research lies in a **novel and efficient mask propagation framework**. This framework encompasses a *potent query-based image segmentor* and a *highly capable query-based flow estimator*, where the whole framework allows for the accurate propagation of mask predictions from key frames to non-key frames.
>
> As discussed in Lines 43-47 and Lines 57-64, applying a strong query-based image segmentor to the task of VSS is highly computational expensive. To reduce the computational cost, current approaches rely on optical flow to propagate features from the key frame to other non-key frames, but they still suffer from performance degradation due to the limitations of pixel-to-pixel optical flow estimation. In this context, the proposed mask propagation framework and the query-based flow module design is non-trivial.
>
> Moreover, the motivation and design of the proposed flow module differ from established models like FlowNet or Transformer-based optical flow models. As discussed in Lines 57-74, traditional optical flow estimation focuses on pixel-level correspondences between adjacent frames. In contrast, our flow module employs learned queries from the key frame to aggregate motion information **at the segment level** for each pair of adjacent frames. This approach captures **segment-specific movement** over time, resulting in enhanced accuracy and consistency for mask propagation in the task of VSS.

---

> > ### Comment · Area_Chair_M63S · 2023-08-16
> > **Reviewer 8jSp**
> >
> > Dear Reviewer 8jSp,
> >
> > Could you please read the author's rebuttal and indicate whether it has changed your opinion?  Currently, you are the only one with a reject rating, so your opinion is very important.  Thank you.
> >
> > Best,
> > AC

---

### Official Review · Reviewer_oxr6 · 2023-07-07

**Soundness:** 3 good
**Presentation:** 3 good
**Contribution:** 3 good
**Rating:** 6
**Confidence:** 4

**Summary:**

This paper investigates an important and fundamental task: video semantic segmentation (VSS). While image semantic segmentation has received significant research attention, VSS has been relatively overlooked due to limited datasets and computational resources. This paper makes a valuable contribution to the field by proposing a method that combines a segmentation network with a flow network. Unlike traditional flow networks, the proposed approach employs query-based flow maps that correspond to each segment. Experimental results demonstrate the effectiveness of the method on two widely-used datasets, namely vspw and cityscape.

**Strengths:**

1. The paper addresses an important but relatively underexplored task.
2. The combination of segmentation and optical flow in a single model is a novel approach, particularly considering the use of query-based flow maps.
3. The proposed method achieves commendable performance on standard datasets such as vspw and cityscape.

**Weaknesses:**

1. The paper lacks specific details regarding the training of the flow modules. For example, what is the loss function used to train query-based flow? How are the flow networks (encoder/decoder) initilized? Do you use trained weights for these modules? Also, more visual examples about the query-based flow maps should be given.
2. The rationale behind utilizing both pixel-wise flow (F^{PF}) and query-based flow (F^{QF}) to generate query-based flow maps is not well-justified. In my opinion, pixel-wise flow is the flow map for every pixel while the query-based flow is only for specific segments. Equation 4, which concatenates both flows, may make the query-based flow not focus on the segments, but on all the pixels.
3. It would be helpful to include information on the number of parameters for all the methods listed in Table 1. Additionally, it would be beneficial to specify the frames per second (fps) achieved by the mask2former and mpvss methods using a single GPU for inference.

**Questions:**

Please see the weakness

**Limitations:**

Please see the weakness

---

> ### Author Rebuttal · Authors · 2023-08-10
>
> Thank you for the constructive comments. We address all the questions below:
>
> **Q1.** Training details for the proposed flow module, e.g., loss function used to train query-based flow, initialization of the flow network (encoder/decoder), utilization of pretrained weights. More visual examples of query-based flow maps should be provided.
>
> **A1.** We have mentioned the training details for the proposed MPVSS which involves the training process for the proposed flow module in Lines 168-171 and Lines 263-264. (1) Loss function: During the training phase, the classification loss and binary mask loss are employed on the class embeddings and warped mask embeddings, respectively, after performing bipartite matching between queries and the ground truth masks, as demonstrated in Mask2Former. Subsequently, the loss gradients are propagated backward across the model to update the proposed flow module. (2) Initialization of the flow network: for the motion encoder, we utilized the weights from FlowNet encoder which is pre-trained on the synthetic Flying Chairs dataset; the motion decoder and flow head are randomly initialized. We will make the training details clearer in the revised version.
>
> We provide more visualization of query-based flow maps in **Figure A** in the general response PDF.
>
>
> **Q2.** Justification of the rationale behind utilizing both pixel-wise flow and query-based flow to generate flow maps.
>
> **A2.** First, as mentioned in lines 237-239, we use the pixel-wise flow to refine the query-based flow and generate the final flow maps for each segment. Second, to better understand the rationale of utilizing pixel-wise flow and query-based flow, we provide visualization for pixel-wise flow and query-based flow separately in **Figure A**. The query-based flow and pixel-wise flow are compensated. The query-based flow captures the overall motion for the specific segment, while the pixel-wise flow adds more detailed movement at the fine-grained level. By fusing the two kinds of flow, the final flow maps are more stable for mask propagation.
>
> **Q3.** Information on the number of parameters and FPS.
>
> **A3.** We add number of parameters and FPS for each method in Table A and Table B in general response. Compared with other methods, the proposed MPVSS achieves both higher FPS and mIoU scores with different backbones on both datasets, which demonstrates the favorable accuracy and efficiency trade-off. We will include Table A and Table B with number of parameters and FPS in the revised version.

---

> > ### Comment · Area_Chair_M63S · 2023-08-18
> > **Reviewer oxr6**
> >
> > Dear Reviewer oxr6,
> >
> > Could you please read the author's rebuttal and other reviews, and indicate whether your comments have been addressed? Thank you.
> >
> > Best, AC

---

> > ### Comment · Reviewer_oxr6 · 2023-08-18
> >
> > Thanks for your response. My concerns have been addressed. I would like to keep my initial rating: weak accept.

---

### Author Rebuttal · Authors · 2023-08-10

We sincerely thank all reviewers for their valuable comments.

### Novelty

Most reviewers recognize the novelty of our method.

*"The combination of segmentation and optical flow in a single model is a novel approach, particularly considering the use of query-based flow maps."* (Reviewer oxr6)

*"The novel query-based flow estimation module introduced in this paper surpasses traditional pixel-wise optical estimation methods..."* (Reviewer 41Ay)

*" The proposed method of propagating the key frame’s query to the motion features is interesting and novel. "* (Reviewer cyWY)

### Promising results

All reviewers agree that our method provides a competitive accuracy and efficiency trade-off.

*"The proposed method achieves commendable performance on standard datasets such as vspw and cityscape."* (Reviewer oxr6)

*"The proposed method demonstrates competitive performance on two widely recognized benchmarks... These results are achieved while significantly reducing computation costs (FLOPs)"* (Reviewer 41Ay)

*"The proposed method achieves SOTA performance with significantly reduced computational cost through extensive experiments on standard benchmarks."* (Reviewer LSXE)

### Accuracy and efficiency trade-off

In Table A and Table B, we add the number of parameters for methods listed in Table 1 and Table 2 of the main paper. FPS is measured on a single NVIDIA V100 GPU with 3 repeated runs. Compared with other methods, the proposed MPVSS achieves favorable accuracy and efficiency trade-off. We will include Table A and Table B with the number of parameters and FPS in the revised version.

Table A. Performance comparisons with state-of-the-art methods on VSPW dataset.

|    Methods     | Backbone | mIoU | WIoU | VC$_8$ | VC$_{16}$ | GFLOPs | #Params |  FPS  |
| :------------: | :------: | ---- | ---- | ---- | ----- | ----- | :-----: | :---: |
| Deeplabv3+[4]  |   R101   | 34.7     | 58.8     | 83.2     | 78.2      |  379.0     |  62.7   | 9.25  |
|  UperNet[56]   |   R101   | 36.5     | 58.6     |  82.6    | 76.1      |  403.6     |  83.2   | 16.05 |
|   PSPNet[71]   |   R101   |  36.5    | 58.1     |  84.2    |  79.6     | 401.8      |  70.5   | 13.84 |
|   OCRNet[65]   |   R101   | 36.7     | 59.2     | 84.0     | 79.0      |  361.7    |  58.1   | 14.39 |
| SegFormer[58]  |  MiT-B2  | 43.9     |  63.7    | 86.0     | 81.2      |  100.8     |  24.8   | 16.16 |
|   SegFormer    |  MiT-B5  | 48.9     | 65.1     |  87.8    | 83.7      | 185.0      |  82.1   | 9.48  |
|  CFFM-VSS[50]  |  MiT-B2  |  44.9    | 64.9     |  89.8    |  85.8     | 143.2      |  26.5   | 10.08 |
|    CFFM-VSS    |  MiT-B5  | 49.3     | 65.8     |  90.8    | 87.1      |  413.5     |  85.5   | 4.58  |
| MRCFA[51]  |  MiT-B2  | 45.3     | 64.7     | 90.3     |  86.2     | 127.9      |  27.3   | 10.7  |
|  MRCFA    |  MiT-B5  | 49.9     | 66.0     |  90.9    |  87.4     |  373.0     |  84.5   | 5.02  |
| Mask2Former[7] |   R50    | 38.5     |  60.2    |  81.3    |  76.4     | 110.6      |  44.0   | 19.44 |
|                |   R101   |  39.3    | 60.2     | 81.3     |  76.4     | 141.3      |  63.0   | 16.90 |
|                |  Swin-T  | 41.2     |  60.1    | 82.5     |  77.6     |  114.4     |  47.4   | 17.13 |
|                |  Swin-S  |  42.1    | 62.6     | 84.5     |  80.0     |  152.2     |  68.9   | 14.52 |
|                |  Swin-B  |  54.1    | 63.1      | 84.7     |  79.3     |  223.5     |  107.1  | 11.45 |
|                |  Swin-L  | 56.1     | 70.3     | 86.6     |  82.9     |  402.7     |  215.1  | 8.41  |
|     **MPVSS**     |   R50    |  37.5    | 59.0  |  84.1  |  77.2     |  38.9     |  84.1   | 33.93 |
|                |   R101   | 38.8     |  59.0    |  84.8    | 79.6      |  45.1     |  103.1  | 32.38 |
|                |  Swin-T  |  39.9    |  62.0    |  85.9    |  80.4     | 39.7     |  114.0  | 32.86 |
|                |  Swin-S  |  40.4    |  62.0    |  86.0    |  80.7     |  47.3     |  108.0  | 30.61 |
|                |  Swin-B  |  52.6    |  68.4    |  89.5    |  85.9     |  61.5     |  147.0  | 27.38 |
|                |  Swin-L  |  53.9    |  69.1    | 89.6     |  85.8     |  97.3     |  255.4  | 23.22 |

Table B. Performance comparisons with the VSS methods on Cityscapes.

| Methods     | Backbone | mIoU | GFLOPs | #Params (M) | FPS   |
| :---------: | :------: | :--: | :----: | :---------: | :---: |
| FCN[37]         | R101     |  76.6    |  2203.3      | 68.5        | 2.83  |
| PSPNet[71]      | R101     | 78.5     |  2048.9      | 67.9        | 2.88  |
| SegFormer[58]   | MiT-B1   | 78.5   |  243.7   | 13.8        | 20.7  |
| SegFormer   | MiT-B5   |  82.4|  1460.4   | 84.7        | 7.20  |
| CFFM-VSS[50]   | MiT-B0   |   74.0   |  80.7      | 4.6         | 15.79 |
| CFFM-VSS    | MiT-B1   |  75.1    |   158.7     | 15.4        | 11.71 |
| MRCFA[51]   | MiT-B0   |   72.8   |  77.5     | 4.2         | 16.55 |
| MRCFA    | MiT-B1   |  75.1    |  145     | 14.9        | 12.97 |
| Mask2Former[7] | R50      | 79.4     |  529.9      | 44.0        | 6.58  |
|             | R101     |  80.1    |   685.5     | 63.0        | 5.68  |
|             | Swin-T   |  82.1    |   543.6     | 47.4        | 5.41  |
|             | Swin-S   |  82.6    |   730.1     | 68.7        | 4.31  |
|             | Swin-B   |  83.3    |   1057.0     | 107.0       | 3.26  |
|             | Swin-L   |  83.3    |   1911.3     | 215.0       | 2.11  |
| **MPVSS**   | R50      | 78.4     |   173.2     | 84.1        | 13.43 |
|             | R101     |  78.2    |  204.3      | 103.1       | 12.55 |
|             | Swin-T   |  80.7    |  175.9      | 114.0       | 12.33 |
|             | Swin-S   |  81.3    |  213.2      | 108.0       | 10.98 |
|             | Swin-B   | 81.7     |   278.6     | 147.0       | 9.54  |
|             | Swin-L   |  81.6    |     449.5   | 255.4       | 7.24  |

---

### Decision · Program_Chairs · 2023-09-21

**Decision:**

Accept (poster)

**Comment:**

This paper proposes a mask propagation framework for efficient video semantic segmentation. The paper received 3 weak accepts, 1 reject, and 1 borderline accept recommendations from reviewers. Positive points included the novelty of the approach, and competitive accuracy and efficiency trade-off. Negative points included lack of motivation/details for some approach components, lacking novelty (especially by one reviewer), unclear performance improvements compared to similar related works, and missing FPS + computational costs. Most of these concerns were adequately addressed by the rebuttal. The reviewer who gave the reject rating did not participate in the post-rebuttal discussions; however, the ACs felt that the reviewer’s major concerns were addressed by the rebuttal. Overall, after carefully considering the paper, rebuttal, and discussions, the ACs feel that the paper makes a good contribution and recommend accept. It is recommended that the authors incorporate the visualizations of the query-based flow and pixel-wise flow into the final supplementary material.